# Benzene Derivatives from Ink Lead to False Positive Results in Neonatal Hyperphenylalaninemia Screening with Ninhydrin Fluorometric Method

**DOI:** 10.3390/ijns6010014

**Published:** 2020-02-29

**Authors:** Shuren Feng, Joanne Mei, Lu Yang, Ping Luo, Xiaonan Wang, Yuan Wang, Jingyi Yao, Lan Cui, Lei Pan, Zefang Wang, Li Xin

**Affiliations:** 1Tianjin Women’s and Children’s Health Centre, Tianjin 300070, China; yaojingyi82@126.com (J.Y.); cuilan68@126.com (L.C.); panlei2121@126.com (L.P.); xinli66@126.com (L.X.); 2Newborn Screening and Molecular Biology Branch, Division of Laboratory Sciences, Centers for Disease Control and Prevention, Atlanta, GA 30341, USA; jvm0@cdc.gov; 3Department of Biology, College of Life Sciences, Tianjin University, Tianjin 300072, China; yanglululu@tju.edu.cn (L.Y.); zefangwang@tju.edu.cn (Z.W.); 4Department of Basic Medicine, Tianjin Medical College, Tianjin 300222, China; luoping62@163.com; 5Tianjin Medical Laboratory, BGI-Tianjin, BGI-Shenzhen, Tianjin 300308, China; douding.wxn@gmail.com; 6Binhai Genomics Institute, BGI-Tianjin, BGI-Shenzhen, Tianjin 300308, China; yuanwang@genomics.cn

**Keywords:** Neonatal Screening, hyperphenylalaninemia (HPA), false-positive, ninhydrin fluorometric test, specimen collection devices, filter paper, newborns, dried blood spots

## Abstract

Ninhydrin-based fluorometric quantification of phenylalanine is one of the most widely used methods for hyperphenylalaninemia (HPA) screening in neonates due to its high sensitivity, high accuracy, and low cost. Here we report an increase of false positive cases in neonatal HPA screening with this method, caused by contamination of blood specimen collection devices during the printing process. Through multiple steps of verification, the contaminants were identified from ink circles printed on the collection devices to indicate the positions and sizes of blood drops. Blood specimens from HPA-negative persons collected on these contaminated collection devices showed positive results in the fluorometric tests, but negative results in tandem mass spectroscopy (MS/MS) experiments. Contaminants on the collection devices could be extracted by 80% ethanol and showed an absorption peak around 245 nm, suggesting that these contaminants may contain benzene derivatives with similar structure to phenylalanine. High-performance liquid chromatography (HPLC) analysis of the ethanol extracts from contaminated collection devices identified two prominent peaks specifically from the devices. Methyl-2-benzoylbenzoate (MBB, CAS#606-28-0) was found as one of the major chemicals from contaminated collection devices. This report aims to remind colleagues in the field of this potential contamination and call for tighter regulation and quality control of specimen collection devices.

## 1. Introduction

Hyperphenylalaninemia (HPA) or more commonly known as phenylketonuria (PKU) is characterized by elevation of blood phenylalanine (Phe) levels [1]. The most deleterious effect of this disease is its irreversible impairment on the developing brains of patients. So far, there has been no permanent treatment to this disease [2,3]. Due to its harmful effects on the developing brain of newborns, early diagnosis and treatment is crucial to the normal development of the patients [4,5].

The methodologies adopted for HPA screening may vary among screening laboratories, but specimen collection devices, consisting of filter paper with preprinted circles used to collect blood and attached to a demographic section, are employed for all these screening methods to obtain dried blood samples (DBS) for new born screening (NBS). Chemical and physical properties of filter paper used to collect DBS have been shown to be important for maintaining stability and shelf life of blood samples [6]. Criteria for the manufacture of specimen collection devices were specified in detail by the Clinical and Laboratory Standards Institute (CLSI NBS01-A6) in regard to the characteristics of materials used for making the devices and performance parameters in order to guarantee reliable test results [7].

In China, the Neonatal Screening Center also requires that blood samples be collected on qualified collection devices, which should contain the same quality of filter paper as the filter paper used for standards and quality controls found in the testing kit [8]. Due to the high price and low market availability of imported 903™ or Ahlstrom 226 filter paper, not all laboratories are using the U.S. Food and Drug Administration licensed medical collection devices for neonatal blood sampling.

Tianjin is one of the four municipalities directly under the central government in China, with a total of about 13 million residents living in 16 county level administrative areas [9]. Funded by Tianjin municipal finances, NBS for congenital hypothyroidism (CH) and HPA is free of charge for all local residents. While the screening is highly recommended for all neonates, it is indeed voluntary as required by the local legislations [8,10]. After birth of their children, all parents are informed of the risks of these diseases and importance of screening procedures. A written acknowledgement is obtained from the parents, either consenting with the screening or acknowledging the risks of forgoing the screening if they decided to do so [9].

In our laboratory, the collection devices originally used imported 903™ filter paper provided by Perkin Elmer, but were discontinued in 2009 due to low market supplies. Specimen collection devices used after 2009 were provided by a third-party manufacturer in China. These collection devices used a general laboratory grade of Whatman filter paper, but the paper was not grade 903™, which is a regulated medical whole blood collection device for neonatal screening. Collection devices that are cleared by the US FDA are made from high-purity cotton linters and manufactured to yield accurate and reproducible blood specimens [7]. Filter papers sold for general use in scientific laboratories are not designed to collect blood nor can they guarantee the same precision and reproducibility required of a medical device for neonatal screening [7].

No problems were found for these collection devices until 2017 when a burst of false positive results were observed for HPA screening by the ninhydrin-based fluorometric method, causing an increased number of infants to be recalled to hospitals for confirmation tests (Figure 1).

Recall rates of newborns for HPA screening due to false positive results in Tianjin were around 0.3% in the past. A large increase of recall rates started in June of 2017, peaked in September, and lasted until early 2018, during which recall rates above 2% were observed in all districts. Starting in February 2017, the increase of false positive results began with two districts, Wuqing District and Xiqing District (data not shown). The fact that early increases of false positive results were observed in only two hospitals misled us, as we focused our investigation on hospital-specific screening procedures. We began our investigation by retesting the specimens with elevated phenylalanine levels and then double-checked our experimental procedures and quality control results. We found our procedures were carried out properly, and all quality control results were in the expected ranges. We then contacted staff from these hospitals to assure their specimen collection procedures and sampling and storage areas did not introduce potential factors that may affect our tests. Multiple aspects of specimen collection procedures were inspected and reinforced accordingly, including the following:(a) if there were changes in transport and storage conditions of specimen collection devices;(b) if there were changes in the care of neonates before blood sampling, for example, feeding special dietary supplements, intravenous infusion of nutritional ingredients, medication, etc.;(c) if there were changes in the environment in which blood specimens were dried, for example, using of chemical reagents for sanitization or air freshener;(d) if there were changes in the conditions for DBS storage before and during transport to the screening laboratory.

In spite of the measures taken above, false positive cases continued from increasingly more hospitals. We still observed about 20 to 40 false positive results in about 1000 DBS specimens from more than ten hospitals each week, even though our experimental conditions were maintained without change, and procedures for specimen collection and testing were carried out according to standardized procedures. When unexpected increases of false positive results were observed in specimens from hospitals in almost all districts, it became clear that something commonly used by all hospitals was related to the phenomenon. We then began to investigate the specimen collection devices issued from our screening centre, used by all hospitals in the municipality.

The increase of false positive results in the screening project means that more families had to undergo unnecessary re-sampling and hospital visits, which may be associated with increased psychological stress in addition to the burden of taking care of neonates. Additionally, this also increased both working load for the screening laboratory and financial burden for the screening program. Here we report our experience to identify the contaminants that caused the increase of false positive results due to the use of filter paper that did not comply with international standards [7].

## 2. Samples and Methods

### 2.1. Samples

Affiliated with Tianjin Women and Children’s Health Center (TWCHC), our screening laboratory receives neonatal DBS on collection devices from the obstetrics and gynecology departments of over 50 secondary or tertiary hospitals scattered in 13 county level administrative areas. From 2009 to 2018, we screened about 80,000 neonates each year. During this period, all collection devices used for the NBS program were provided by Perkin Elmer (FI-20750, Turku, Finland), who purchased them from a third-party company named Cangzhou Yikang Food and Drug Packaging Company Limited (Cangzhou, China). After discovering that the devices were responsible for the increase in false positive results, collection devices from a different company, Guizhou Liding Biotechnology Limited Liability Company (Guiyang, China), were purchased and dispatched to replace the old ones. Before replacing the old devices, new collection devices were tested for contamination in our laboratory. The sample collection and research protocol reported in this work were evaluated and approved by the Ethics Committee for Clinical Research of TWCHC in October, 2017 (approval code: kylw191220).

### 2.2. Screening Procedures

Screening of HPA was conducted using the Neonatal PHENYLALANINE quantification kit from LabSystems Diagnostics, Oy (REF199896, FIN-01720 Vantaa, Finland). All procedures were conducted following the manufacture’s protocol, which was “based on the modification of the McCaman and Robins quantitative fluorometric method [11] modified for dried blood spots and adapted to a microplate fluorometer with high throughput [12,13,14]”, as described in the protocol. Specimens containing blood phenylalanine concentrations above 109 µmol/L (1.8 mg/dL) in the initial tests were retested in duplicate. Retested specimens with phenylalanine concentrations between 120 µmol/L (2 mg/dL) and 210 µmol/L (3.5 mg/dL) required recall of the patients to the nearest county level Women and Children’s Health Centres (WCHCs) for DBS recollection and retesting. Patients with phenylalanine concentrations equal to or above 210 µmol/L were referred to our screening centre for confirmatory tests and initiation of dietary treatments if necessary. Phenylalanine concentrations above 120 µmol/L for confirmatory tests were considered positive cases and subjected to further diagnostic examinations and long-term follow up under a clinician’s guidance. Phenylalanine concentrations below 120 µmol/L for confirmatory tests were considered false positive cases. In this report, recall rates caused by false positive cases were calculated by dividing total number of neonates screened in the laboratory with total number of false positive cases.

### 2.3. Fluorometric Tests of DBS Collection Devices for Contaminants

DBS collection devices showing false positive results in the fluorometric tests were randomly selected for background fluorescence tests. Discs of 3 mm in diameter were punched from the collection devices from areas without DBS and then tested using the phenylalanine quantification kit following standard procedures for HPA screening as described above.

New collection devices (Lot20180180) from Guizhou Liding Biotechnology Limited Liability Company were tested in the laboratory for background signals using the fluorometric method. Filter paper discs of 3 mm in diameter were punched from the collection devices from both within the pre-printed ink circle and areas outside the circle, and tested for fluorescence with the phenylalanine quantification kit. Then, new collection devices were randomly sampled, attached with old collection devices, and provided to selected hospitals so that neonatal DBS could be collected on both types of collection devices. It should be emphasized that same number of DBS will be collected as before for these collection devices with additional filter paper attached, but one of the DBS was to be placed on the attached new filter paper. Specimen collection devices were sent back to the laboratory and tested following standard procedures. After making sure that the new collection devices had no contamination issues, they were dispatched to county level WCHCs, which distributed new collection devices to all affiliated hospitals to replace old collection devices. It took more time than had been anticipated for the old collection devices to be totally replaced by the new collection devices in local hospitals, which means that even after knowing the cause of false positive results, there were still old collection devices being used for neonatal blood sampling for a while.

### 2.4. Tandem Mass Spectrometry

Blood spots of 3.2 mm in diameter punched from DBS collection devices were extracted at 30 °C for 30 min with 90 µL methanol including isotopic tracers from Cambridge Isotope Laboratories Inc. (Tewksbury, MA, USA). After centrifugation at 2000× *g* for 5 min, 50 µL supernatant was transferred to another 96-well plate and nitrogen-dried at room temperature. Followed by addition of 50 µL derivatization reagent, the sample plate was incubated at 60 °C for 30 min and then nitrogen-dried. After reconstitution with 75 µL of 80% acetonitrile in H2O (*v*/*v*) and centrifugation at 4000× *g* for 15 min, 10 µL of the supernatant was taken for liquid chromatographic mass spectrometric (LC-MS) analysis.

The Waters Acquity UPLC I-Class/Xevo TQD tandem mass spectrometer (Milford, MA, USA) was used to measure the concentration levels of multiple amino acids and acylcarnitines by the application of direct flow-injection analysis and multiple reaction monitoring (MRM). The mass spectrometer was tuned with resolution, capillary voltage, source temperature, and desolvation temperature set as 0.7 Da, 3.5 kV, 120 °C, and 350 °C, respectively. The supernatant was introduced into the electrospray ionization interface by the mobile phase with scheduled flow rates. All the targeted analytes were quantitatively measured according to the ion pair transitions of both their own and corresponding internal standards. Waters Masslynx NT 4.1 Software was used to control the UPLC-MS/MS as well as data processing and analysis.

### 2.5. UV–Visible Spectrometry

In order to identify the chemical properties of contaminants on the suspected collection devices, ten discs of 3 mm in diameter were punched off from either on the ink circles or inside the ink circles for both suspected (Lot160001) and new collection devices (Lot20180180) and soaked in 400 µL of 80% ethanol at 23 °C for 35 min in four test tubes, respectively. The extraction solutions were transferred to new test tubes and centrifuged at 16,000× *g* for 15 min at 23 °C, after which the supernatants were carefully transferred to new test tubes. UV–Visible spectra of the above samples were obtained at room temperature by a NanoDropTM 2000 spectrophotometer from Thermo Scientific (Wilmington, DE, USA). Blank measurements were taken using 80% ethanol, and all samples were measured afterwards with five technical repeats. Final spectra of each sample were obtained by averaging all five spectra for each sample. 

### 2.6. HPLC

White paper discs of 3 mm in diameter were punched from unused collection devices either on the ink circles or inside the ink circles for both suspected collection devices (Lot160001) and new collection devices (Lot20180180), and soaked in 1 mL of 80% ethanol at 23 °C for 35 min in four test tubes, respectively. The extraction solutions were transferred to new test tubes and centrifuged at 16,000× *g*, 23 °C for 10 min. The supernatants were then taken into new tubes and centrifuged at 16,000× *g*, 23 °C for 20 min, after which supernatants were transferred into new test tubes. Mobile phase solutions were prepared with ultra-pure water and chromatography-grade ethanol. All mobile phase solutions and collection device extraction samples were degassed and passed through 0.22 µL organic solvent resistant filters before being analysed by HPLC with the procedure described below on Waters Alliance E2695 Separations Module equipped with a UV detector that measured absorption at 215 nm.

Briefly, 100µL of solutions from above were loaded on to 10% ethanol pre-equilibrated high-purity reversed-phase C18 column (column length: 250mm, inner diameter: 4.6 mm, column volume: 5 mL, particle size: 5 µm in diameter) from Thermo Scientific The column was then washed by a linear gradient of 10% to 40% ethanol with a total volume of 15mL at flow rate of 0.5 mL per minute. The elution was then conducted at the same flow rate with a linear gradient of 40% to 95% ethanol with a total volume of 35 mL, which was then followed by a 10mL linear gradient of 95% to 100% ethanol. The column was finally washed by a linear gradient of 100% to 10% ethanol. The temperature for the HPLC experiment was maintained at 23 °C.

### 2.7. Gas Chromatography–Mass Spectrometry (GC-MS)

In order to have an adequate amount of extract for GC-MS analysis, 600 filter discs of 3 mm in diameter were punched from ink circles of contaminated collection devices, Lot160001, and soaked in 4 mL of 80% ethanol for 35 min at 23 °C. The extraction solution was centrifuged and filtered with the same procedures as samples for HPLC analysis and subjected to GC-MS analysis. Ethanol solution used for contaminant extraction was also tested as control. Excess amount of anhydrous sodium sulphate was added to sample solutions to remove residual water before mass spectral analysis.

Samples were analysed using a Thermo Fisher GCMS ISQ LT mass spectrometer. Gas chromatography was conducted at constant helium flow at 1 mL per minute using polysiloxane with 5% phenyl substitutions as stationary phase. Resolving temperature was started at 60 °C for 5 min, then increased to 100 °C at a rate of 3.5 °C per minute, maintained for 5 min, then increased to 200 °C at 8 °C per minute, maintained for 5 min, then increased to 300 °C at 15 °C per minute, and kept for 15 min. 

The mass spectrometer was tuned with electron ionization power source voltage, ion source temperature, capillary temperature, injection valve temperature, and scanning spectrum set as 70 eV, 320 °C, 280 °C, 280 °C, and 50–650 *m*/*z*, respectively. Fractions resolved by gas chromatography were injected to mass spectrometer at 280 °C with scheduled flow rate. Mass spectra for all fractions were compared against the NIST mass spectral database (developed by the National Institute of Standards and Technology, Gaithersburg, MD, USA, and provided along with the spectrometer software) using the spectrometer software to obtain the chemical identity of each component [15]. Relative fraction amounts identified from spectral comparison were calculated by normalization of areas under each peak against total area under all peaks.

## 3. Results

### 3.1. Increased False Positive Cases Were Due to Contamination of Specimen Collection Devices by Chemicals Other Than Phenylalanine

We started by testing the background fluorescence of specimen collection devices from which false positive results were observed during HPA screening. White filter paper discs (3 mm in diameter) were punched from collection devices adjacent to DBS with high phenylalanine values and subjected to the fluorometric screening tests. Discs from collection devices from which negative screening results were observed from the same hospital(s) were also randomly chosen for comparison. 

As shown in Figure 2A, white discs from DBS with either false positive or negative results all displayed high background fluorescence. Specifically, increased background fluorescence readings were found in specimen collection devices of two lot numbers (Lot160001 and Lot6944444/W444). This indicated that the collection devices were contaminated and likely caused the observed false positive results.

However, it was not clear if the contaminant(s) was (were) phenylalanine, or something that may react with the test reagents in the screening assay, or something that may be intrinsically fluorescent. Thus, we used a MS/MS technique that directly measures the concentration of phenylalanine based on its mass and charge ratio in the testing system. As can be seen from Figure 2B, all samples with false positive results in the fluorometric tests gave negative results in the MS/MS experiments, whereas control samples from HPA patients with known concentrations of phenylalanine gave phenylalanine concentrations in expected ranges. Considering that the MS/MS method quantifies phenylalanine in blood samples by specifically testing the position and intensities of phenylalanine on the mass/charge spectrum and their ratio to that of isotopically labelled phenylalanine of known concentrations, this result clearly showed that the contaminants were unlikely to be phenylalanine. Once it was known that the collection devices were contaminated, we immediately informed our supplier of this issue and requested changing of the specimen collection device. 

### 3.2. Higher Concentration of Contaminants Was Observed from Ink Circles on the Collection Devices

The next question was to identify the contaminant(s) and how they contaminated the collection devices. We discussed this situation with colleagues from another screening laboratory in Tianjin and experts from Beijing Neonatal Screening Centre, which also used the fluorometric methods for HPA screening. They had both experienced unexpected increases of false positive cases in the past, but with less severity and for a shorter period. A piece of valuable information was provided by Beijing’s expert Dr. Ma Zhijun: they used to test their collection devices for background fluorescence and found that filter paper discs from the ink circles had higher background fluorescence than that of discs punched from areas without ink. Thus, we decided to test our specimen collection devices (Lot160001 and Lot6944444/W444) and new collection devices from a different manufacturer (Lot20180180) to see if contaminants could be identified from ink circles of these collection devices.

As can be seen from Figure 3A, background fluorescence intensities from contaminated collection devices were much higher than that of new collection devices (Lot20180180) and uncontaminated collection devices (Lot6231706/41). Furthermore, fluorescence intensities of discs from the ink ring on contaminated collection devices were significantly higher than that of discs from areas within the pre-printed ring without ink. In comparison, the background fluorescence intensities of discs from uncontaminated collection devices that included a portion of the ink ring or within the pre-printed circle were in the same range, without significant difference.

This result clearly showed that the ink-covered areas contained higher concentrations of contaminant(s) than areas without ink on contaminated collection devices, suggesting that the ink used for these collection devices might contain components that were intrinsically fluorescent or reacted with testing reagents.

To make sure those new collection devices were free of contaminants that may interfere with the test before using them to replace old collection devices, uncontaminated collection devices (Lot20180180) were attached to old collection devices and used for neonatal blood sampling in selected hospitals. As can be seen from Figure 3B, fluorescence intensities of blood samples from the same baby collected on new collection devices (▲) fell within the range below the cut-off value (○) and above background values of white discs from the new collection devices (■); whereas larger numbers of samples with fluorescence intensities higher than the cut-off value were observed for the same set of blood spots collected on contaminated collection devices (*), whose background fluorescence (×) was more variable. Interestingly, fluorescence values of white filter paper discs from some contaminated collection devices (×) were much higher than the cutoff fluorescence value (○) and fluorescence values of blood samples collected on the same collection devices (▲).

This suggests that the contaminants on the collection devices may not have an additive effect on the quantification results of phenylalanine in blood samples with this method, which is consistent with the MS/MS results that excluded the possibility of the contaminant being phenylalanine. Thus, it can be inferred that fluorescent signals observed for samples from contaminated collection devices were not specifically from phenylalanine and may originate from other contaminants.

### 3.3. Absorption Peak around 245 nm Was Only Observed on Samples from Contaminated Collection Devices, Which May Be Represented by Two Elution Peaks Specific for Contaminated Collection Devices on HPLC Chromatograms

The ninhydrin-based fluorometric screening cannot differentiate phenylalanine from chemicals with similar structures. Previous studies showed that formation of fluorophores in this test involves two steps: first, phenylalanine is converted to phenylacetaldehyde through oxidation and decarboxylation; second, phenylacetaldehyde combines with primary amines from the dipeptide in the presence of ninhydrin to form the fluorophore that is excitable at 390 nm and emits at 480 nm [16]. Thus, if the ink used for printing the circles on the collection devices contained chemicals that can be oxidized into phenylacetaldehyde or similar compounds, an increase of fluorescent signal in HPA screening would likely be expected. 

In order to verify if the ink circles printed on contaminated collection devices contained chemicals that may be able to form fluorophores in the screening experiment, filter paper discs were punched from the ink circle and extracted by 80% ethanol. Figure 4A shows that chemicals extracted from the contaminated collection devices contained an absorption peak at around 245 nm and no absorption in the visible range of wavelength. Since the excitation wavelength of our fluorescence assay was 390 nm, absence of an absorption peak at this wavelength in the paper disc extracts indicated that the fluorescence observed for false positive cases in the screening experiments was not directly from the contaminant but from the fluorophores formed by the contaminants with the reaction mixtures. HPLC analysis of the ink circle extracts identified two prominent peaks specific to the contaminated collection devices (Figure 4B), which were present both on ink circles with higher concentrations and from areas without ink. Due to the low concentration of contaminants in fractions corresponding to the two elution peaks, further analysis of the contaminants by nuclear magnetic resonance spectroscopy was not practical.

### 3.4. Major Contaminants from Ink Circles Were Identified by GC-MS 

GC-MS analysis of filter paper discs from contaminated collection devices showed that the major contaminant extracted by 80% ethanol was methyl-2-benzoylbenzoate (MBB, CAS 606-28-0). Table 1 lists all chemical compounds and their relative proportions identified by GC-MS from either the background solvent (80% ethanol) or extracted contaminants. 

Only three chemicals were found in both samples by GC-MS, shown as bold typeface CAS registry numbers. Chemicals with at least one benzene ring in their structures (underlined CAS numbers) were found to be the major components of contaminants. No benzene ring derivatives were found in the control sample. Methyl-2-benzoylbenzoate, also referred to as photoinitiator OMBB, is widely used in a variety of food packaging, UV ink, and varnish systems. The decomposition product of this chemical under our experimental conditions with the ninhydrin method may include benzaldehyde, only one carbon less than the phenylacetaldehyde that is involved in the formation of fluorophore in the HPA screening experiment. Considering its wide application in printing industries, it was not surprising to identify this chemical on the ink circle from the contaminated collection devices.

## 4. Conclusions

We identified a serious contaminant in whole blood collection cards used for newborn screening. The contaminated collection cards caused a high recall rate for HPA among our screened population of newborns. Some important conclusions can be drawn from this study.

Because of a lack of availability of filter paper approved by FDA (or another such regulatory organization), the filter paper used to manufacture specimen collection devices purchased by the neonatal screening centre did not comply with the international standard, CLSI NBS01 [8]. The benzene derivatives in the ink on these devices reacted with our ninhydrin assay used to test for phenylalanine, and they directly lead to the high recall rate, causing undue anxiety for the families who were asked to bring their newborns back for additional testing. The additional work needed to investigate the high recall rate added to the cost borne by the laboratory for reagents, supplies, repeat testing, follow up, and staff time.

The US Centers for Disease Control and Prevention (CDC) annually publishes their results from independent testing of all FDA-approved filter paper for whole blood collection for newborn screening programs [17]. CDC voluntarily evaluates statistical samples of filter paper, before it is printed into collection devices for newborn screening programs for serum volume per 3.2 mm punch, absorption time (sec), and spot diameter. The performances of new lots of approved filter paper are evaluated against established criteria for these parameters. CLSI NBS01 [7] is used by filter paper manufacturers to assure their devices conform to established specifications and includes criteria for ink. Specifically, CLSI NBS01 states, “The supplier and manufacturer must assure that the printing ink will not interfere for all NBS assays” [7]. Validation data should be available from the paper manufacturer upon request. 

We recommend that if a newborn screening program does not have access to approved filter paper for their collection devices, they must establish protocols to test the devices before they are used by hospitals. At a minimum, if the manufacturer cannot provide data showing the ink is inert, the program, prior to distribution to hospitals and birthing facilities, should test areas of printed and unprinted collection devices to assure the ink has no contamination or interference with the newborn screening assays.

Newborn screening is expanding into areas where either limited screening has been conducted or where it is being established for the first time. The international standard, CLSI NBS01 [7], has been in existence for decades and is updated periodically. Newborn Screening programs should use this standard for best practices regarding the manufacture of whole blood collection devices, collection of good-quality DBS specimens, and storage and transport of devices from hospitals or collection sites to screening laboratories.

Often, the only time a heel stick specimen can be collected is soon after birth and before the newborn is discharged from the birthing facility. It is incumbent on the newborn screening program to obtain the best quality specimen on collection devices that meet established performance criteria and provide samples free of contaminants. The success of newborn screening systems to identify newborns at risk of disease is dependent on the specimen collection device. 

## Figures and Tables

**Figure 1 IJNS-06-00014-f001:**
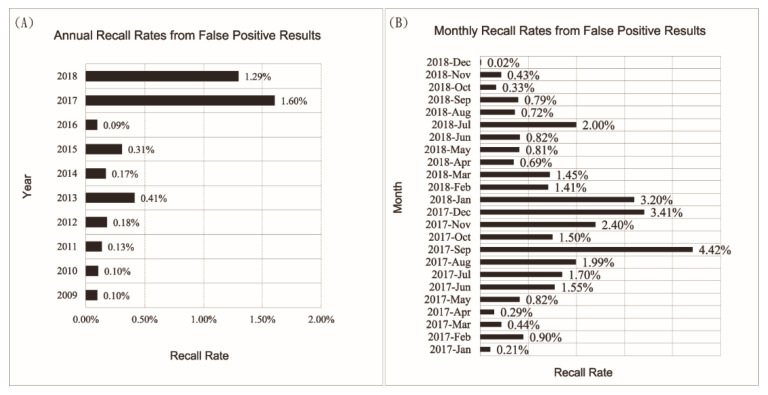
Recall rates of neonatal hyperphenylalaninemia (HPA) screening resulting from false positive results in Tianjin, China. (**A**) Annual recall rates due to false positive cases, numbers calculated by dividing total number of infants screened each year with number of infants recalled to the laboratory whose confirmatory tests were negative. (**B**) Monthly recall rate for 2017 and 2018 calculated in the same way, showing the gradual decrease of recall rate after the contaminated collection devices were replaced with new collection devices starting in January of 2018.

**Figure 2 IJNS-06-00014-f002:**
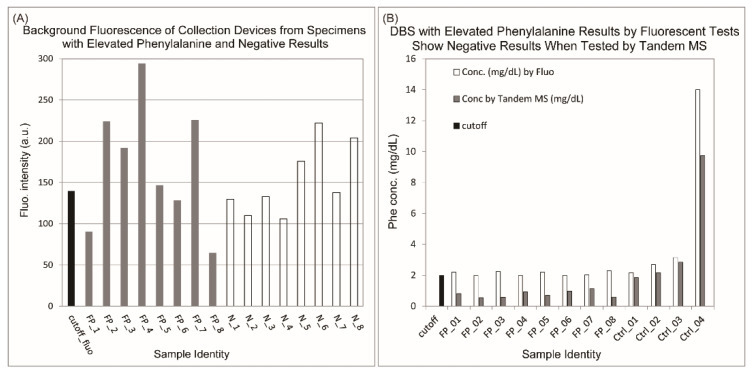
Increased false positive cases due to contamination of specimen collection devices by chemicals other than phenylalanine. (**A**) Fluorescent intensities above the cutoff value were observed for white filter paper discs from randomly selected samples with either false positive (FP) or negative results (N) from the same hospital. (**B**) Samples with false positive results in fluorescent tests showed negative results when tested by the tandem MS method.

**Figure 3 IJNS-06-00014-f003:**
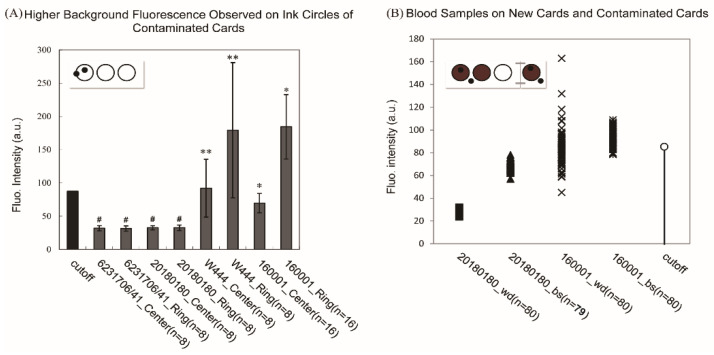
The difference between new collection devices and contaminated collection devices were identified by fluorescent tests. (**A**) White filter paper discs were punched from unused collection devices either on the ink circle or within the circle for background tests; and locations of discs on the collections devices are shown by round black dots either on the ink ring or inside the ring on the inset diagram. Contaminated collection devices showed higher fluorescence background than new collection devices, and fluorescence intensities of paper discs from the ink circle (ring) are significantly higher than that of discs from the center area (center) without ink. Two-tailed unpaired t-tests were conducted for the differential analyses. (#, *p* > 0.7; **, *p* = 0.010; *, *p* < 0.001). Error bars were 99.5% confidential intervals around the mean value of fluorescence intensities. (**B**) Single pieces of new card (Lot20180180, right section of the card with one ink circle) were attached to contaminated collection devices (Lot160001, left part of the card with three ink circles) and used for blood sampling as shown by the inset diagram. Punch locations were shown as round black dots. Blood discs (bs) and white filter discs (wd) from both old (* and × are data lables for bs and wd samples, respectively) and new collection devices (▲ and ■ are data lables for bs and wd samples) were tested. In the 20180180 bs data set (*n* = 79), in one of 80 cases, the attached new card was not used by the individual collecting the blood sample in the hospital.

**Figure 4 IJNS-06-00014-f004:**
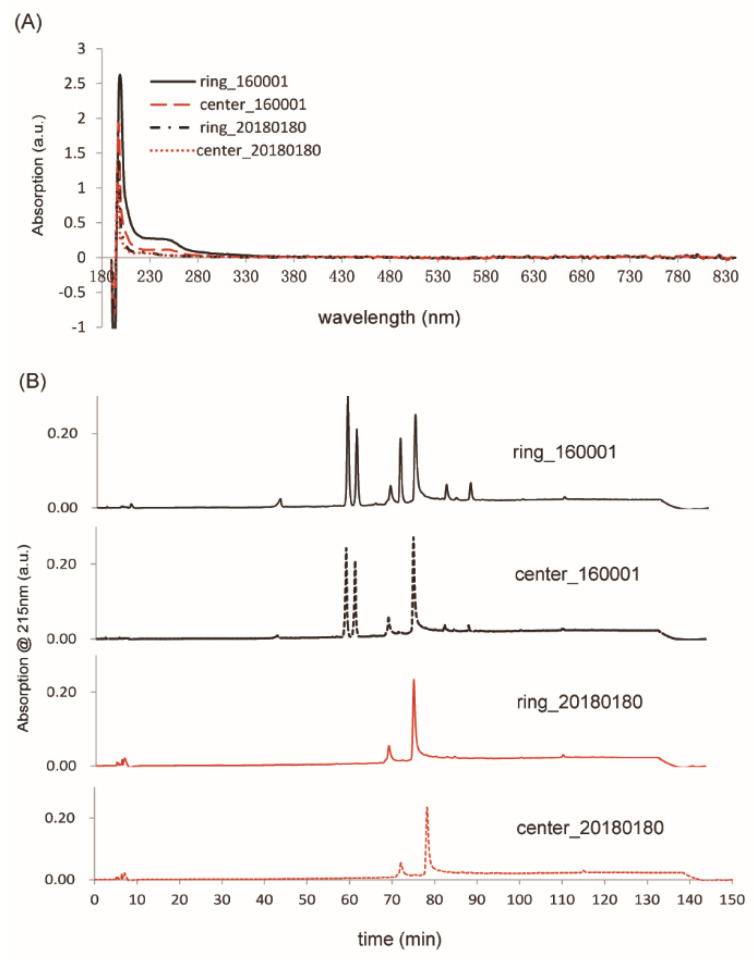
UV–Visible absorption spectroscopic and HPLC analysis of contaminants extracted from collection devices. (**A**) UV–Visible spectra of filter paper discs punched on the ink circle (ring) or inside the ink circle (center) from both contaminated (Lot160001) and new (Lot20180180) collection devices. All extractions have maximum absorption at around 200 nm. Extractions of contaminated collection devices have an additional absorption peak around 245 nm. No absorption was observed in the visible wavelength range for all samples. (**B**) HPLC chromatograms of above samples. Two prominent peaks were observed on the chromatograms of samples extracted from contaminated collection devices around 60min elution time, which were absent on samples from new collection devices.

**Table 1 IJNS-06-00014-t001:** GC-MS analysis of chemicals from ink circles on contaminated specimen collection devices extracted by 80% ethanol. Chemicals found in both samples were shown as bold typeface CAS numbers; chemicals with benzene ring in their structures were shown as underlined CAS numbers.

Background (80% Ethanol)		Extracted Contaminants	
CAS#	Portion(%)	CAS#	Portion(%)
538-24-9	36.56	606-28-0	38.6
6114-18-7	32.42	106-90-1	11.94
**111-61-5**	8.75	57-10-3	8.34
**77-90-7**	8.53	57-11-1	6.87
**628-97-7**	3.37	2156-97-0	6.7
6848-50-0	2.89	**77-90-7**	6.61
7098-22-8	2.35	13048-33-4	4.52
7459-33-8	1.82	629-96-9	4.06
1560-84-5	1.51	**111-61-5**	3.97
1560-96-9	1.42	96-76-4	3.45
1560-88-9	1.38	**628-97-7**	2.33
-- --	-- --	112-80-1	1.24
-- --	-- --	119-61-9	0.72
-- --	-- --	137-89-3	0.49

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
