# Peer review of "Benzene Derivatives from Ink Lead to False Positive Results in Neonatal Hyperphenylalaninemia Screening with Ninhydrin Fluorometric Method"

_2409-515X, 2020, doi:10.3390/ijns6010014_

Round 1
Reviewer 1 Report
This is an interesting and important paper for newborn screening, and complements /supports the standard for blood collection paper and cards CLSI NBS01. However, it is much too long and the key message is lost in peripheral detail. I suggest a major rewrite along the following lines -
Introduction - screening in the region, how we found the problem (some of this is in methods). Significantly shorten by eliminating repetition and omitting details not relevant to the cards and printing eg details of treatment for PKU. Test for PKU belongs here too. Method - how the cause of the problem of false positive tests was investigated (it was the cards, and the printing). Details of the analysis are done well. Results - it was the ink (#3.1 etc should be in the introduction as part of how the problem was identified)
Please don't refer to blood collection paper as filter paper, it isn't - and although the screening community know what is meant this isn't universal.
Author Response
Reviewer 1
This is an interesting and important paper for newborn screening, and complements /supports the standard for blood collection paper and cards CLSI NBS01. However, it is much too long and the key message is lost in peripheral detail. I suggest a major rewrite along the following lines -
Introduction - screening in the region, how we found the problem (some of this is in methods). Significantly shorten by eliminating repetition and omitting details not relevant to the cards and printing eg details of treatment for PKU. Test for PKU belongs here too. Method - how the cause of the problem of false positive tests was investigated (it was the cards, and the printing). Details of the analysis are done well. Results - it was the ink (#3.1 etc should be in the introduction as part of how the problem was identified)
Our responses and revisions (shown by underlined texts in revised manuscript):
We agree with the reviewer’s suggestions to focus on the core question in the article: the cards and printing. Thus we have rewritten the manuscript by:
1) Eliminating detailed contents not relevant to cards and printing, including details of treatment for PKU in the first paragraph.
2) Moving section3.1 to Introduction section as part of the initial investigation as requested by the reviewer.
We tried not to disrupt the logic flow of this investigation when rewriting the manuscript. Further suggestions and comments will be greatly appreciated.
Please don't refer to blood collection paper as filter paper, it isn't - and although the screening community know what is meant this isn't universal.
Our responses and revisions:
Firstly, we totally agree with Reviewer1 in the sense that we should be careful to avoid confusions when referring different part of the collection devices. Thus we reviewed the manuscript from the beginning, and changed the “filter paper” phrase into “collection devices” where appropriate, shown by “track changes” records in line51, 52, 69, and 115 of revised manuscript.
Secondly, we respectfully disagree with Reviewer 1. “Filter paper” is a legitimate term and has been used in the title of the CLSI standard, NBS01, since its inception decades ago and for every version thereafter. NBS01 is in its final revisions for version 7. The term “filter paper” refers to the product used to collect blood. The term “collection device” refers to both the filter paper with preprinted circles and to the demographic portion, to which the filter paper is glued. It is the filter paper, with preprinted circles, that was the issue in this study, and not the whole collection device. Please see updated text with “track changes” in section 4. Conclusions.
The “collection device” is both the filter paper and the demographic part that is filled out when the heel stick is collected. From the latest version of NBS01 (to be published April 2020): “collection device//specimen collection device – For purposes of this standard, a medical device used to collect blood spots used for routine newborn screening; NOTE 1: The collection device has two components: a section for recording demographic and other requested information and a filter paper section with preprinted circles to be filled with the newborn blood drops; NOTE 2: If a preprinted circle is not present, local requirements must define the quantity of blood considered acceptable; NOTE 3: Once the blood is collected, the collection device becomes the specimen (also referred to as blood spot specimen or dried blood spot specimen) and is no longer considered a collection device; NOTE 4: This specimen collection device is also commonly referred to as “filter paper” or a blood spot “card”.”
CDC voluntarily evaluates FILTER PAPER before is it made into a collection device.
Reviewer 2 Report
A clearly written and interesting "detective story" including the resolution of the mystery.
Author Response
A clearly written and interesting "detective story" including the resolution of the mystery.
Our responses and revisions (shown by underlined texts in revised manuscript):
We are grateful for the reviewer’s encouraging comments. No change seemed to be necessary at this moment.
Reviewer 3 Report
Feng et al. present a thorough evaluation of false positive newborn screening results for hyperphenylalaninemia due to a common chemical used in printing, especially in ultraviolet-fixed inks.
The steps in the investigation and the conclusion are clearly expressed.
There are a few correctable errors which would strengthen this manuscript.
On the line 84, the use of reference 12 is incorrect as Gerasimova and co-workers wrote nothing about the filter paper cleared by the United States FDA. The use of reference 12 on line 86 is appropriate as this is a conclusion of that Gerasimova et al. paper.
There is an additional 'i' included in the Diagnostics of the corporate name.
Line 113 starts with a typographical error and with a reviewers query. Changing the first word to 'manufacturer's' resolves the typographical issue. If LabSystems Diagnostics specifically cites the 2 referenced papers for the method for use of their kit, then a clarification phrase indicating that the protocol was derived from those 2 papers should be included.
Line 135 appears to have a typographical error. If the intent of the authors is to convey that the same amount of blood (number of blood spots) were collected but one was to be placed on the second lot of filter paper, a minor rewrite as needed.
In line 165, the word "contaminated" is a conclusion rather than indicating 'suspect' lot. It is also the first time that the lot number is identified.
Line 175 again uses 'contaminated' where an alternative may be more appropriate when referring to the tested material in the method section rather than as a conclusion. The use of the word 'contaminated' in the conclusions is justified by the significant amount of work outlined in the results section of the paper.
A simplification of language on line 201 is possible. Perhaps re-word to: Samples were analyzed using a Thermo Fisher...
The superscript denoting the reference is incorrect on line 212.
On line 243, the words "and have been" should be eliminated. If additional clarity is necessary, a phrase behind the word hospitals, to say 'in the municipality' could be added.
The figure 3B includes the misleading data point in the 20180180bs data set that results from selecting a point from the contaminated card. Since this point does not represent a sample on the new collection devices, it should removed from the plot and the N of samples decreased to 79 accordingly. The statement in the legend can be re-written to indicate that, in 1 of 80 cases, the attached new card was not used by the individual collecting the sample.
Lines 324-326 read as a net conclusion of the investigation rather than as documentation of the result.
On line 362 space is needed after Table.
On line 367, rather than 'bolder', 'bold typeface' remote use of potential ambiguity as the registry does not include bolder and less bold chemicals. These lines also referred to the CAS number where in the table both the number and the decimal fraction, in the background analysis, are in bold face.
On line 390, the number for the citation needs to be corrected.
On line 445 the reference is incorrect. This cited paper does not describe newborn screening, filter paper use or specimen properties. Perhaps the authors intended to use a different reference.
In line 457 there are 2 typographical errors. The first is stylus take as both feet and and S should be behind the surname. Furthermore the journal needs to be cited. This particular paper is from Clinical Chemistry.
Author Response
Reviewer 3
Feng et al. present a thorough evaluation of false positive newborn screening results for hyperphenylalaninemia due to a common chemical used in printing, especially in ultraviolet-fixed inks.
The steps in the investigation and the conclusion are clearly expressed.
There are a few correctable errors which would strengthen this manuscript.
On the line 84, the use of reference 12 is incorrect as Gerasimova and co-workers wrote nothing about the filter paper cleared by the United States FDA. The use of reference 12 on line 86 is appropriate as this is a conclusion of that Gerasimova et al. paper.
Our responses and revisions (shown by underlined texts in revised manuscript):
In the revised manuscript, reference by Gerasimova and co-workers is not cited after this sentence (line70 in revised manuscript ), only CLSI NBS01-A6 is cited.
There is an additional 'i' included in the Diagnostics of the corporate name.
Our responses and revisions (shown by underlined texts in revised manuscript):
Corrected accordingly on line 134 in the revised manuscript.
Line 113 starts with a typographical error and with a reviewers query. Changing the first word to 'manufacturer's' resolves the typographical issue. If LabSystems Diagnostics specifically cites the 2 referenced papers for the method for use of their kit, then a clarification phrase indicating that the protocol was derived from those 2 papers should be included.
Our responses and revisions (shown by underlined texts in revised manuscript):
The typographical error was corrected accordingly. The sentence has been rewritten to clarify the reviewer’s concerns. (line134-137 in revised manuscript)
Line 135 appears to have a typographical error. If the intent of the authors is to convey that the same amount of blood (number of blood spots) were collected but one was to be placed on the second lot of filter paper, a minor rewrite as needed.
Our responses and revisions (shown by underlined texts in revised manuscript):
The typographical error was corrected accordingly. The sentence has been rewritten as requested. (line 159-162)
In line 165, the word "contaminated" is a conclusion rather than indicating 'suspect' lot. It is also the first time that the lot number is identified.
Line 175 again uses 'contaminated' where an alternative may be more appropriate when referring to the tested material in the method section rather than as a conclusion. The use of the word 'contaminated' in the conclusions is justified by the significant amount of work outlined in the results section of the paper.
Our responses and revisions (shown by underlined texts in revised manuscript):
We agree with the reviewer’s opinion. We have changed “contaminated” into “suspected” in these two places. (line 189, 191, 201 in revised manuscripts)
A simplification of language on line 201 is possible. Perhaps re-word to: Samples were analyzed using a Thermo Fisher...
Our responses and revisions (shown by underlined texts in revised manuscript):
We have rewritten the sentence as suggested. (line 227, 228 in revised manuscript)
The superscript denoting the reference is incorrect on line 212.
Our responses and revisions (shown by underlined texts in revised manuscript):
This was a typographical error in the original manuscript. It has been corrected in revised manuscript. (line238 in revised manuscript)
On line 243, the words "and have been" should be eliminated. If additional clarity is necessary, a phrase behind the word hospitals, to say 'in the municipality' could be added.
Our responses and revisions (shown by underlined texts in revised manuscript):
We have revised this sentence accordingly. This section was moved to Introduction as suggested by Reviewer 1. (line109 in revised manuscript)
The figure 3B includes the misleading data point in the 20180180bs data set that results from selecting a point from the contaminated card. Since this point does not represent a sample on the new collection devices, it should removed from the plot and the N of samples decreased to 79 accordingly. The statement in the legend can be re-written to indicate that, in 1 of 80 cases, the attached new card was not used by the individual collecting the sample.
Our responses and revisions (shown by underlined texts in revised manuscript):
We have revised both the figure and legend as suggested. The revised figure was also provided separately with the revised manuscript. (line308-309 in revised manusript)
Lines 324-326 read as a net conclusion of the investigation rather than as documentation of the result.
Our responses and revisions (shown by underlined texts in revised manuscript):
We have revised this sentence accordingly. (line324-326 in revised manusript)
On line 362 space is needed after Table.
On line 367, rather than 'bolder', 'bold typeface' remote use of potential ambiguity as the registry does not include bolder and less bold chemicals. These lines also referred to the CAS number where in the table both the number and the decimal fraction, in the background analysis, are in bold face.
Our responses and revisions (shown by underlined texts in revised manuscript):
We have corrected the typographical errors accordingly. (line363 and table 1 in revised manuscript)
On line 390, the number for the citation needs to be corrected.
Our responses and revisions (shown by underlined texts in revised manuscript):
We have corrected the typographical errors accordingly. (line 391 in revised manuscript)
On line 445 the reference is incorrect. This cited paper does not describe newborn screening, filter paper use or specimen properties. Perhaps the authors intended to use a different reference.
Our responses and revisions (shown by underlined texts in revised manuscript):
This was a mistake when the author was formatting the references using Endnote while opening multiple word documents and reference libraries. The reviewer was correct, we meant to cite the work by Chace DH et al, 2015. (line447, Reference 6 in revised manuscript)
In line 457 there are 2 typographical errors. The first is stylus take as both feet and and S should be behind the surname. Furthermore the journal needs to be cited. This particular paper is from Clinical Chemistry.
Our responses and revisions (shown by underlined texts in revised manuscript):
The error has been corrected and missing information renewed in revised manuscript. (line462, reference12 in revised manuscript)
Round 2
Reviewer 1 Report
Thank-you for considering the comments made in the initial review.
Author Response
Dear Dr. Fingerhut and Dr. Schielen,
Thank you for giving us the opportunity to make additional revisions. We would like to respond to comments regarding the terms “filter paper” and “collection device”.
- Reviewer comments stating “special blood collection paper should not be called filter paper”:
- The CLSI international standard NBS01 Blood Collection on Filter Paper for Newborn Screening Programs has been in existence since 1982. Throughout the 38-year history of the document and its seven revisions, the title has not changed. The Document Development Committee (DDC) for the current revision discussed the risks and benefits to creating an alternative term for “filter paper”, however because of the deep historical context of the document and the need for consistency, the DDC rejected changing the title and terminology therein. This decision was upheld by the larger CLSI Expert Panel on Newborn Screening. A better approach is to allow more manuscripts describing issues encountered with blood collection devices that do not comply with international standards. Publication of this manuscript offers IJNS an opportunity to educate the newborn screening community on the importance of ensuring that manufacturers of filter paper follow NBS01 and provide data showing equivalency to the standard, and companies that prepare collection devices using filter paper also follow NBS01 and do not introduce substances that interfere with newborn screening tests.
- The ISNS lexicon of terms is dated and attempts by the organization to update terminology have not been successful. ISNS has, therefore, opted to follow and support the CLSI effort to harmonize terms in their newborn screening documents. The US Association of Public Health Laboratories’ Newborn Screening and Genetic Testing Committee and the Quality Assurance/Quality Control Subcommittee are in the process of reviewing all CLSI newborn screening terms. The committees will prepare a report and recommendations to CLSI. Changing “filter paper” or “collection device” to something else is not part of the discussion.
- NBS01, version 7, which should be published in the spring of 2020, gives the following definition for collection device:
collection device//specimen collection device (for newborn screening) – For purposes of this standard, a medical device used to collect blood spots used for routine newborn screening; NOTE 1: The collection device has two components: a section for recording demographic and other requested information and a filter paper section with preprinted circles to be filled with the newborn blood drops; NOTE 2: If a preprinted circle is not present, local requirements must define the quantity of blood considered acceptable; NOTE 3: Once the blood is collected, the collection device becomes the specimen (also referred to as blood spot specimen or dried blood spot specimen) and is no longer considered a collection device; NOTE 4: This specimen collection device is also commonly referred to as “filter paper” or a blood spot “card”.
We have tried to adhere to this terminology within the manuscript.
- Academic Reviewer comments
NBS01, version 7, defines "collection device (for newborn screening)" as the filter paper attached to the demographic information. For FDA and other regulatory purposes, the term "device" is needed to denote that it is used to collected human blood. In this manuscript, the filter paper was contaminated in the process of creating the collection device.
At the request of a previous reviewer, changes were made to the last revision that may have created some confusion. I would like to suggest the following changes (see attached manuscript containing the edits listed below):
- Lines 42-44, Page 1: "collection device" is defined where it first appears in the manuscript
Change the sentence to: "The methodologies adopted for HPA screening may vary among screening laboratories, but specimen collection devices, consisting of filter paper with preprinted circles used to collect blood and attached to a demographic section, are used for all these screening methods to obtain dried blood samples (DBS) for NBS.”
- Lines 46, Page 2: replace "devices" with "filter paper" in "Chemical and physical properties of filter paper used to collect DBS have been shown to be important for maintaining stability and shelf life of blood samples6."
- Lines 52-53, Page 2: replace "collection devices" with "filter paper" in "In China, the neonatal screening center also requires that blood samples be collected on qualified collection devices, which should contain the same quality of filter paper as the filter paper used for standards and quality controls found in the testing kit8.”
- Lines 119-121, Page 3: replace "collection devices" with "filter paper" in "Here we report our experience to identify the contaminants that caused the increase of false positive results, due to the use of filter paper that does not comply with international standards7.
We hope these responses satisfy the Reviewer and Academic Reviewer’s comments.